# Re-randomized Densification for One Permutation Hashing and Bin-wise Consistent Weighted Sampling

**Ping Li**
Cognitive Computing Lab
Baidu Research
Bellevue, WA 98004, USA
liping11@baidu.com

**Xiaoyun Li**[*]
Department of Statistics
Rutgers University
Piscataway, NJ 08854, USA
xiaoyun.li@rutgers.edu

**Cun-Hui Zhang**[†]
Department of Statistics
Rutgers University
Piscataway, NJ 08854, USA
cunhui@stat.rutgers.edu

## Abstract

Jaccard similarity is widely used as a distance measure in many machine learning and search applications. Typically, hashing methods are essential for the use of Jaccard similarity to be practical in large-scale settings. For hashing binary (0/1) data, the idea of one permutation hashing (OPH) with densification significantly accelerates traditional minwise hashing algorithms while providing unbiased and accurate estimates. In this paper, we propose a "re-randomization" strategy in the process of densification and we show that it achieves the smallest variance among existing densification schemes. The success of this idea inspires us to generalize one permutation hashing to weighted (non-binary) data, resulting in the so-called "bin-wise consistent weighted sampling (BCWS)" algorithm. We analyze the behavior of BCWS and compare it with a recent alternative. Experiments on a range of datasets and tasks confirm the effectiveness of proposed methods. We expect that BCWS will be adopted in practice for training kernel machines and fast similarity search.

## 1 Introduction

In recent years, there has been a surge of interest in studying the following measure of similarity for nonnegative data [17, 6, 12, 26, 14, 20, 29, 21, 22]:

$$J(S,T) = \frac{\sum_{i=1}^{D} \min(S_i, T_i)}{\sum_{i=1}^{D} \max(S_i, T_i)} \tag{1}$$

where $S, T \in \mathbb{R}^D$ are two $D$-dimensional data vectors with only nonnegative entries. This "min-max" measure is a generalization of the "Jaccard similarity" in binary (0/1) data. For simplicity, in this paper, we will use "Jaccard" regardless whether the data are binary or non-binary. We should also mention that $J(S,T)$ has been successfully extended to include data with negative entries [21, 22]. In fact, under a fairly general distributional assumption, $J \to \frac{1-\sqrt{(1-\rho)/2}}{1+\sqrt{(1-\rho)/2}}$, where $\rho$ is the correlation [25].

While $J(S,T)$ in Eq. (1) appears deceivingly simple, the work of [20, 21, 22] demonstrated that, through extensive empirical studies, this measure of similarity is surprisingly effective when it is used as a kernel for classification (e.g., SVM and logistic regression). In many public datasets, using this (tuning-free) kernel resulted in a substantial increase in classification accuracy, compared to the (best-tuned) radial basis function (RBF) kernel. Furthermore, the "tunable" version [22] of $J(S,T)$ is even able to achieve classification accuracy comparable to boosted trees (and deep nets) [18, 19].

Since $J(S,T)$ is a type of nonlinear kernels. In order to use it for mere medium-scale datasets, we must be able to "linearize" this kernel. Scaling nonlinear kernel machines is a known non-trivial task [2]. For example, we cannot even store a kernel matrix in the memory for a dataset with only 1,000,000 training samples, which has $10^{12} \approx 2^{40}$ entries and will need multiple terabytes of storage.

---

[*]The work of Xiaoyun Li was conducted during the internship at Baidu Research.

[†]The work of Cun-Hui Zhang was conducted as a consulting researcher at Baidu Research.

## 1.1 Consistent Weighted Sampling (CWS)

The method of consistent weighted sampling (CWS) [12, 26, 14] is the standard strategy for efficiently computing the Jaccard similarity in Eq. (1). This algorithm is summarized in Algorithm 1, for hashing the vector $S$ as an example. For all other data vectors (e.g., $T$), we apply the same randomization, i.e., using the same random numbers $(r_i, c_i, \beta_i)$ in Algorithm 1. For vectors $S$ and $T$, we denote the outputs as $(i_S^*, t_S^*)$ and $(i_T^*, t_T^*)$, respectively. Then the following interesting probability result holds:

$$\mathbf{Pr}\left(i_S^* = i_T^* \ \text{ and } \ t_S^* = t_T^*\right) = J(S, T) \tag{2}$$

---

**Algorithm 1:** Consistent Weighted Sampling (CWS).

---
1 **Input:** (Non-negative) Data vector $S_i, \ i = 1$ to $D$
2 **Output:** Consistent uniform sample $(i^*, t^*)$

3 For every nonzero $S_i$
4    $r_i \sim Gamma(2, 1), \ c_i \sim Gamma(2, 1), \ \beta_i \sim Uniform(0, 1)$
5    $t_i \leftarrow \lfloor \frac{\log S_i}{r_i} + \beta_i \rfloor, \ \ a_i \leftarrow \log(c_i) - r_i(t_i + 1 - \beta_i)$
6 End For
7 $i^* \leftarrow arg \min_i \ a_i, \qquad t^* \leftarrow t_{i^*}$

---

After repeating the randomization $M$ times, one can then estimate the similarity as

$$\hat{J} = \frac{1}{M} \sum_{j=1}^{M} 1\left\{i_{S,j}^* = i_{T,j}^* \ \text{ and } \ t_{S,j}^* = t_{T,j}^*\right\} \tag{3}$$

$$\mathbb{E}(\hat{J}) = J, \qquad Var(\hat{J}) = \frac{1}{M}J(1 - J) \tag{4}$$

Note that this estimate $\hat{J}$ is actually a linear (inner product) kernel in a (sparse) high-dimensional space. Basically there will be only exactly $M$ 1's if we expand the samples into one sparse vector. The prior work [20, 21, 22] already demonstrated the effectiveness of CWS for training kernel SVMs.

## 1.2 The Computational Bottleneck of CWS

CWS as presented in Algorithm 1 is fairly complex. Also, it needs $O(\bar{f}M)$ computations to process one data vector, where $\bar{f}$ is the average number of nonzero entries in the data vector. For many important applications, $\bar{f} \ll D$, especially when $D$ is large (i.e., high-dimensional data). This cost is actually very expensive and can be the bottleneck if engineers hope to use CWS in practice. It would be highly desirable if the computational cost (per data vector) can be reduced to $O(\bar{f})$. Note that since we anyway have to touch each data entry (at least) once, $O(\bar{f})$ is the minimal possible cost.

## 1.3 Bin-wise CWS (BCWS)

Bin-wise CWS (or BCWS) appears to be a natural idea, although it has taken us a fairly long journey to get it into the current form as presented in this paper. We will elaborate on the algorithm in details and explain other attempts we had tried which did not lead to satisfactory results.

With BCWS, we first conduct a random permutation on the columns of the data matrix, then we group the columns into equal-sized bins and perform CWS on each bin separately. Suppose we break the columns into $K = M$ bins and each bin contributes one sample, then the processing cost would be only $O(\bar{f})$ (where $\bar{f}$ is the average number of nonzero entries). Intuitively, this strategy should work well if $D$ is large and the data entries are "well-behaved" (e.g., entries follow a Gaussian or more generally a non-heavy-tailed distribution). This is because, when $D$ is large and $K$ is not too large, each bin with $D/K$ entries would still be a good representative sample of the original data.

The real world is typically not this ideal. When $D$ is large, the real data often tend to be (highly) sparse, which means some bins may have only a small number of nonzero entries or even empty. Therefore, we must be able to deal with empty bins. Algorithm 2 is a generic description of BCWS.

---
**Algorithm 2:** A generic description of BCWS.
---
1 Randomly group a dataset of $D$ columns evenly into $K$ bins.
2 For each data vector.
3     For $j = 1$ to $M$
4         Pick one non-empty bin.
5         Generate one CWS sample.     */\* In fact, we can replace CWS with other methods, e.g., [6, 29]. \*/*
6     End For
7 End For

---

This is just the beginning of the story. Next, we need to develop strategies to implement lines 4 and 5.

1. ***How to pick a non-empty bin?*** We present the results on two strategies.
   (a) The first strategy is to treat, for each data vector, the bins as a $K$-dimensional binary data vector and apply classical min-wise hashing [4, 3] to choose a non-empty bin. We denote this strategy as **Rs** (random select).
   (b) A better strategy is to apply the idea of "one permutation hashing" [24] and densification [30]. Instead of always (randomly) selecting a non-empty bin, we focus only on the empty bins. If a bin is empty, then we select a bin from the non-empty ones according to some strategy. It was shown in [30] that a good strategy is to fully randomly select from all bins and stop till a non-empty bin is reached. We denote it as **Den** (densification) and will describe this strategy in more details.
2. ***How to generate a CWS sample from a non-empty bin?*** We also present two strategies.
   (a) For one non-empty bin, we generate one CWS sample and always output this sample whenever this bin is picked.
   (b) We always generate a new CWS sample whenever this bin is picked.

Therefore, we will present in total four variants of BCWS. Quite a few years back, when we started to study this problem, we had tried various other proposals for picking the non-empty bins. One intuitive strategy, which initially appeared very reasonable, is to pick the bins with probabilities proportional to the sums of the elements in all bins. However, after many unsuccessful attempts, we have eventually realized that we should just focus on whether bins are empty or non-empty, i.e., our Algorithm 2.

Note that, for binary data, the CWS algorithm generates statistically equivalent samples as the classical minwise hashing. Thus, to use BCWS on binary data, we just need to apply the standard minwise hashing method (instead of CWS), whenever a non-empty bin is picked. In other words, for binary data, our study of BCWS actually leads to a new densification scheme. In the next two sections, after we first review minwise hashing, one permutation hashing, and densification, we will illustrate how our work actually improves over the existing densification schemes for binary data.

## 2   Minwise Hashing, One Permutation Hashing, and Densification

Minwise hashing (minhash) [4, 3] was initially developed for efficiently computing data Jaccard similarities (a.k.a. resemblances) for the task of duplicate web page removal. Then the technique has been widely applied for numerous practical tasks, e.g., [10, 28, 11, 7, 8, 13, 9, 27, 16, 32, 15, 5, 1].

Consider two sets $S, T \subseteq \Omega = \{1, 2, 3, ..., D\}$. Suppose a random permutation $\pi$ is performed on $\Omega$: $\pi : \Omega \longrightarrow \Omega$. An elementary probability argument shows that

$$\mathbf{Pr}\left(\min(\pi(S)) = \min(\pi(T))\right) = \frac{|S \cap T|}{|S \cup T|} = J(S, T). \tag{5}$$

For the sake of simplicity, with a slight abuse of notation, we use $J(S, T)$ to denote the Jaccard similarity between two sets $S$ and $T$, and $J(S, T)$ to denote the generalized Jaccard similarity between two vectors $S$ and $T$. In order to estimate $J$, one will need to repeat the permutations $M$ times. This computational burden has been resolved by the idea of "one permutation hashing" [24]. As illustrated in Figure 1, after applying one permutation on the columns, we break the column space evenly into $K$ bins (where $K = M$ in [24]) and then use the locations of the first nonzero entries in all bins as the hashed data. This substantially reduces the processing time and makes minwise hashing truly practical. However, new problems arise as there will inevitably be empty bins in sparse data.

To deal with empty bins, [31] proposed a strategy by directly borrowing hashed data from neighboring bins. The work [30] proposed an improvement by selecting non-empty bins randomly from all bins. Basically, we can understand the strategy in [30] as using a random permutation $\pi' : \{1, 2, ..., K\} \longrightarrow \{1, 2, ..., K\}$. If one bin is empty, one then follows the permutation $\pi'$ till one non-empty bin is found. [30] also used an argument based on "universal hashing" to claim that cycling can be avoided.

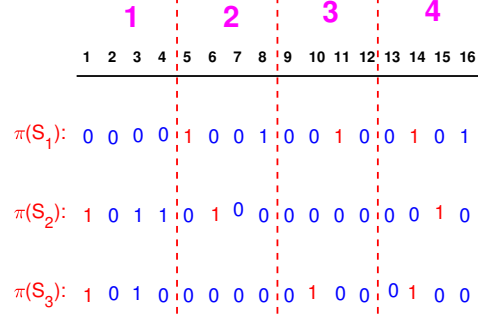

Through an elegant analysis, [30] claimed that it achieved the smallest (optimal) variance. Interestingly, the performance of the densification scheme in [30] can still be improved, in the sense that,

Figure 1: Demonstration of one permutation hashing. For example, for $S_1$, the hashed values are $[*, 5, 11, 14]$ with the first bin being empty.

given a fixed budget of storage (i.e., sample size $M$), the variance can be further reduced via re-randomization on each selected non-empty bin. This finding is useful, practically and theoretically.

## 3  Binary Data: Densification with Re-randomization

We continue to use the example in Figure 1. After one permutation and binning, the non-empty bins for each set constitute a set: $I_{S_1} = \{1, 2, 3\}$ for $S_1$, $I_{S_2} = \{1, 2, 4\}$ for $S_2$, and $I_{S_3} = \{1, 3, 4\}$ for $S_3$, respectively. As briefly mentioned in Introduction, there are four strategies to generate samples. Suppose we first group the data matrix columns into $K$ bins and we hope to obtain $M$ samples.

1. **Rs**: For the original set $S_i$, we randomly select a non-empty bin using minwise hashing on the set $I_{S_i}$. For each non-empty bin, we have already generated a hashed value from minhash within the bin. Once a non-empty bin is selected, we return its hashed value.

2. **RsRe**: For the original set $S_i$, we randomly select a non-empty bin using minwise hashing on the set $I_{S_i}$. Once a non-empty bin is selected, we perform a minwise hashing within the bin and return the hashed value.

   For **Rs** and **RsRe**, we repeat the procedure $M$ times to obtain $M$ samples, regardless of $K$, the number of bins. It is often the case that we use $M = K$, but we do not have to.

3. **Den**: After we group the data matrix columns into $K$ bins, we "densify" (fill in) the empty bins from the beginning of the bins using the "optimal" densification strategy described in [30]. For each non-empty bin, we have already generated a sample by a minwise hashing within the bin. If $M < K$, then we stop till we have collected $M$ samples.

4. **DenRe (Densification with Re-randomization)**: For each empty bin, after we fill it in with one non-empty bin, we re-do a minwise hashing within that bin and output a sample.

Here we will repeat the procedure for "optimal densification" as described in the nice work of [30]. Conceptually, we have another random permutation $\pi' : \{1, 2, ..., K\} \longrightarrow \{1, 2, ..., K\}$. If one bin is empty, one then follows the permutation $\pi'$ till a non-empty bin is found. Based on the "universal hashing" argument, [30] claimed that cycling can be avoided. The actual implementation is slightly more sophisticated than just using a random permutation. We refer readers to [30] for more details.

Note that, for binary data, **Den** is basically the scheme in [30]. Even though it is called "optimal densification" in [30], in this paper we will show that the variance can actually be further reduced by a re-randomization (**Re**) step, through a careful analysis. Also, note that while [24, 31, 30] always let $M = K$, in this study we have relaxed this constraint by providing more general theoretical results.

### 3.1  Theoretical Results

In the original minwise hashing, we have an (unbiased) estimator $\hat{J}$ of the Jaccard similarity (analogous to Eq. (3)) with a variance $\frac{1}{M}J(1 - J)$ in order to generate $M$ samples. With the four densification schemes described above, we now have four more (unbiased) estimators, which are respectively denoted by $\hat{J}^M_{Rs}$, $\hat{J}^M_{RsRe}$, $\hat{J}^M_{Den}$, and $\hat{J}^M_{DenRe}$. Here, we use the superscript $M$ to emphasize the sample size. The following lemma gives an important quantity in the re-randomization process.

**Lemma 1.** *Let $d = \frac{D}{K}$ be an integer. Assume $B$ is the index set of a simultaneously non-empty bin. Let $f = |S \cup T|$. Denote $\tilde{f} = \sum_{i \in I_B} \max(S_i, T_i)$ the number of nonzeros in bin $B$. Then we have*

$$E_0 \triangleq \mathbb{E}[\frac{1}{\tilde{f}}|I_{emp,B} \neq 0] = \frac{\sum_{j=\max(1,d+f-D)}^{\min(d,f)} \frac{1}{j}\binom{f}{j}\binom{D-f}{d-j}}{\binom{D}{d} - I_{\{d+f-D\leq 0\}}\binom{D-f}{d}}, \tag{6}$$

*where $I_{emp,B}$ denotes the indicator of event that bin $B$ is empty. Conditional on the event that there are $m$ simultaneously non-empty bins, we have*

$$\tilde{E}_0(m) \triangleq \mathbb{E}[\frac{1}{\tilde{f}}|I_{emp,B} \neq 0, m] = \sum_{j=1\vee[f-(m-1)d]}^{d\wedge(f-m+1)} \frac{1}{j}\frac{\binom{d}{j}H(m-1,f-j|d)}{H(m,f|s)}, \tag{7}$$

*where the following recursion holds for $\forall k \leq K$,*

$$H(k,n|d) = \sum_{j=\max\{1,n-(k-1)d\}}^{\min\{d,n-k+1\}} \binom{d}{j}H(k-1,n-j|d), \quad H(1,n|d) = \binom{d}{n}.$$

A careful analysis derives the following theory for the variances of the proposed four estimators.

**Theorem 1.** *Let $\tilde{E}_0(m)$ be defined in Lemma 1. Let $f_1 = |S|$, $f_2 = |T|$, $a = |S \cap T|$, $f = |S \cup T| = f_1 + f_2 - a$, and $J = \frac{a}{f}$, $\tilde{J} = \frac{a-1}{f-1}$. $N_{emp}^M$ is the number of empty bins out of first $M \leq K$ bins. If $M > K$, then $N_{emp}^M = N_{emp}^K$. We have*

$$Var\left(\hat{J}_{Rs}^M\right) = \frac{J}{M} + \frac{M-1}{M}E_1 - J^2,$$

$$Var\left(\hat{J}_{RsRe}^M\right) = \frac{J}{M} + \frac{M-1}{M}E_2 - J^2.$$

*If $M \leq K$, then*

$$Var\left(\hat{J}_{Den}^M\right) = \frac{J}{M} + \frac{1}{M^2}\mathbb{E}[(M - N_{emp}^M)(M - N_{emp}^M - 1)J\tilde{J}] + \frac{1}{M^2}\mathbb{E}[N_{emp}^M(2M - N_{emp}^M - 1)]E_1 - J^2,$$

$$Var\left(\hat{J}_{DenRe}^M\right) = \frac{J}{M} + \frac{1}{M^2}\mathbb{E}[(M - N_{emp}^M)(M - N_{emp}^M - 1)J\tilde{J}] + \frac{1}{M^2}\mathbb{E}[N_{emp}^M(2M - N_{emp}^M - 1)]E_2 - J^2,$$

*If $M > K$, then*

$$Var\left(\hat{J}_{Den}^M\right) = \frac{1}{M^2}[K^2(Var\left(\hat{J}_{Den}^K\right) + J^2) + (M-K)(M+K-1)E_1 + (M-K)J] - J^2,$$

$$Var\left(\hat{J}_{DenRe}^M\right) = \frac{1}{M^2}[K^2(Var\left(\hat{J}_{DenRe}^K\right) + J^2) + (M-K)(M+K-1)E_2 + (M-K)J] - J^2,$$

*where $E_1 = \mathbb{E}[\frac{J}{K-N_{emp}^K} + (1 - \frac{1}{K-N_{emp}^K})J\tilde{J}]$ and $E_2 = \mathbb{E}[\frac{\tilde{E}_0(K-N_{emp}^K)}{K-N_{emp}^K}J + (1 - \frac{\tilde{E}_0(K-N_{emp}^K)}{K-N_{emp}^K})J\tilde{J}]$.*

Note that the above theorem involves the probability distribution of $N_{emp}^M$, where $M \leq K$. [24] derived this probability for a simpler case. Here, we provide the general result.

**Theorem 2.** *The distribution of $N_{emp}^M$, where $M \leq K$, is given by*

$$Pr\left\{N_{emp}^M = j\right\} = \sum_{\ell=0}^{M-j}(-1)^\ell\binom{M}{j}\binom{M-j}{\ell}\binom{D(1-(j+\ell)/K)}{f}\Big/\binom{D}{f}.$$

We also have the asymptotic convergence as the following Theorem.

**Theorem 3.** *Suppose $K$ is finite and fixed, then as $M \to \infty$,*

$$\lim_{M\to\infty} Var\left(\hat{J}_{Den}^M\right) = \lim_{M\to\infty} Var\left(\hat{J}_{Rs}^M\right) = E_1 - J^2,$$

$$\lim_{M\to\infty} Var\left(\hat{J}_{DenRe}^M\right) = \lim_{M\to\infty} Var\left(\hat{J}_{RsRe}^M\right) = E_2 - J^2.$$

*Proof.* The results simply follows by taking $M \to \infty$ in Theorem 1. $\square$

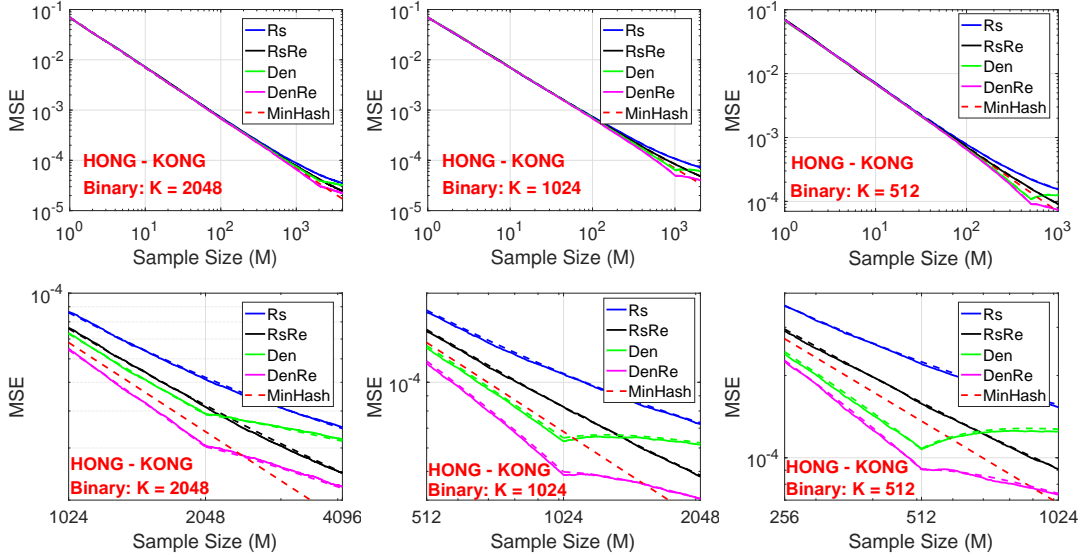

Figure 2: Verification of theoretical results in Theorem 1 for estimating Jaccard similarity between two binary vectors ("HONG" and "KONG"). We report the empirical MSEs (solid curves), which overlap the theoretical variances (dashed curves). Note that it is possible **DenRe** can have smaller variance than the original minwise hashing. (Bottom panels are zoomed-in versions of upper panels.)

## 3.2 Sanity Check: An Empirical Study to Verify the Theoretical Results

Figure 2 presents a sanity check of our theoretical results, from $10^5$ simulations, for estimating the Jaccard similarity between two word vectors: "HONG" and "KONG". Basically, "HONG" denotes the vector of occurrence (0/1) of the word "HONG" in a repository of $D = 2^{16}$ documents. As expected, the data are highly sparse and reflect the real-world situation. This dataset (named "Words") has been used in a few previous papers on hashing and sketching, as early as in 2005 [23]. From Figure 2, we can see that the empirical MSEs (mean square error ($= Var + Bias^2$), solid curves) overlap the (dashed) theoretical curves very well, confirming the theoretical results in Theorem 1.

In Theorem 1, since $E_0 \leq 1$ and $\tilde{J} < J$, we always have $E_2 \leq E_1$ and thus $Var(\hat{J}^M_{DenRe}) \leq Var(\hat{J}^M_{Den})$ and $Var(\hat{J}^M_{RsRe}) \leq Var(\hat{J}^M_{Rs})$ always hold for positive $M$. We can see that the equality is attained only when $|S \cup T| = 1$, i.e., each data vector contains at most 1 nonzero entry. Hence, in essentially all cases, the variance of re-randomized approaches (**RsRe, DenRe**) is smaller than that of the corresponding counterparts (**Rs, Den**), and the improvement can be substantial in some cases.

In fact, this re-randomized densification procedure achieves the smallest variance among all existing OPH variants, since maximum randomness is introduced in both bin selection and hash reassignment steps. Thus, our proposed **DenRe** approach is able to generate the most accurate estimator under the efficient OPH scheme. We also remark that under the setting by [30] where asymptotic analysis is in the sense of $K = M \to \infty$, the variance of $\hat{J}^M_{DenRe}$ is always the smallest and converges to zero.

**Running time.** Let $\bar{f}$ be the average number of nonzero entries of all sets. To generate $M$ samples for each set, the vanilla minwise hashing algorithm has a running time of $O(M\bar{f})$. Clearly, one permutation hashing + densification schemes is able to dramatically reduce the processing time. For simplicity, consider $M = K$. The running time of **Den** scheme is $O(\bar{f} + 2K + \frac{K}{K - N^K_{emp}} N^K_{emp})$. The re-randomized approach **DenRe** takes $O(\bar{f} + 2K + \frac{K}{K - N^K_{emp}} N^K_{emp} + \frac{\bar{f}}{K} N^K_{emp})$, where the additional cost comes from re-doing minwise hashing for each empty bin. This additional cost $\frac{\bar{f}}{K} N^K_{emp}$ is in general minimal. One can also see that, as the number of empty bins $N^K_{emp}$ becomes larger, the variance reduction effect due to re-randomization would be even more substantial.

Next, we will generalize our densification scheme to non-binary data, i.e., the BCWS algorithm.

# 4 Weighted data: Bin-wise Consistent Weighted Sampling (BCWS)

We have provided a comprehensive analysis on densification schemes for one permutation hashing. However, these methods are constrained to binary data. Given the significant acceleration of bin-wise type algorithms, one may ask: can we extend one permutation hashing, which is designed specifically for binary sets, to weighted sets? The main concern is that, unlike in the binary case, different weights are assigned to entries which intrinsically give bins different amount of information. Using the same strategy as in one permutation hashing could no longer provide unbiasedness. In the next, we show that applying bin-wise CWS is also theoretically plausible with moderate number of bins $K$, and it provides significant speedup and very similar empirical performance to the original CWS procedure.

## 4.1 Concentration of BCWS Estimates

Consider two non-negative real-valued data vectors $S, T \in \mathbb{R}^D$. For simplicity, we assume $M = K$. Intuitively, one may expect that the BCWS estimator is "nearly unbiased" if the data is "reasonably behaved", in the sense that information is approximately uniformly distributed among the entries.

**Theorem 4.** *Consider two non-negative real-valued vectors $S, T \in \mathbb{R}^D$. Denote $\mu_1$, $\mu_2$ and $\mu_3$ as the average of $S$, $T$ and $S \vee T$, and $\sigma_1$, $\sigma_2$ and $\sigma_3$ be corresponding standard deviations. Further assume that $\frac{D}{K}$ is an integer. After applying random permutation $\pi$, $h_k(\cdot)$ is the hash tuple generated in bin $k$. Denote $\hat{J}_{BCWS}(\pi) = \frac{1}{K}\sum_{k=1}^K I\{h_k(S) = h_k(T)|\pi\}$ the estimator given by BCWS. For any $t > 0$, when $K \leq \min_i \frac{\mu_i}{\sigma_i}\sqrt{\frac{D}{t}}$, we have*

$$P\left\{\frac{1 - K(\delta_1(t) \vee \delta_2(t))}{1 + K\delta_3(t)}J - \frac{K(\delta_3(t) + (\delta_1(t) \vee \delta_2(t)))}{1 + K\delta_3(t)} \leq \mathbb{E}[\hat{J}_{BCWS}(\pi)]\right.$$

$$\left. \leq \frac{1 + K(\delta_1(t) \vee \delta_2(t))}{1 - K\delta_3(t)}J + \frac{K(\delta_3(t) + (\delta_1(t) \vee \delta_2(t)))}{1 - K\delta_3(t)}\right\} \geq 1 - \frac{6K}{p_1}e^{-t} + \frac{3p_0 K}{p_1},$$

*with $\delta_i(t) = \frac{\sigma_i}{\mu_i}\sqrt{\frac{t}{D}}$ for $i = 1, 2, 3$, $p_0 = \binom{D-f}{D/K}/\binom{D}{D/K}$, $p_1 = 1 - p_0$, and $f = |\{i : S_i > 0 \text{ or } T_i > 0\}|$.*

## 4.2 Experiment: MSEs for Estimating Jaccard in Real-valued Word-Vectors

We again use the word occurrence data (such as "HONG" and "KONG") form the "Words" dataset. This time, we record the actual numbers of occurrences instead of just the presence/absence information. We have conducted a large number of simulations for estimating the Jaccard similarity between two vectors. The patterns are essentially similar and hence we only present the results for three word-vector-pairs, in Figure 3. More details about the datasets and the experiments are available in the supplementary material, which also contains the proofs of the theorems presented in this paper.

As expected, as shown in Figure 3, **DenRe** outperforms **Den** (and two other estimates) in terms of MSEs. We remark that when $K$ is smaller, the curves of CWS, **Den** and **DenRe** are usually indistinguishable (if $M < K$) and hence we just report results with relatively larger $K$ values.

## 4.3 The "R-G" Algorithm for Estimating Jaccard

To demonstrate the advantage of our proposed method for estimating Jaccard similarity with nonnegative real-valued data, here we introduce another interesting algorithm [17, 6, 29] which in this paper we refer to as the "R-G" method. They showed that when data are dense, "R-G" speeds up CWS, typically by a substantial factor.

Figure 4 is an example where two vectors $S = [S_1, S_2, S_3, S_4]$ and $T = [T_1, T_2, 0, T_4]$. The algorithm needs prior fixed feature-wise upper bound $m_i$, $i = 1, ..., D$. The green region represents the data entries. Denote $\tilde{m}_j = \sum_{i=1}^j m_i$. They repeatedly choose a point at random on $[0, \tilde{m}_D]$ until it falls into the green region. The hash values are set to be the number of tries before success. This simple strategy also yields an unbiased estimator of Jaccard

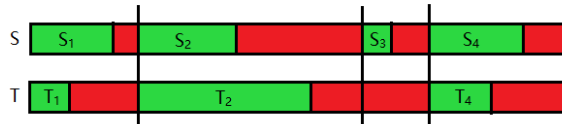

Figure 4: Illustration of R-G algorithm in [29].

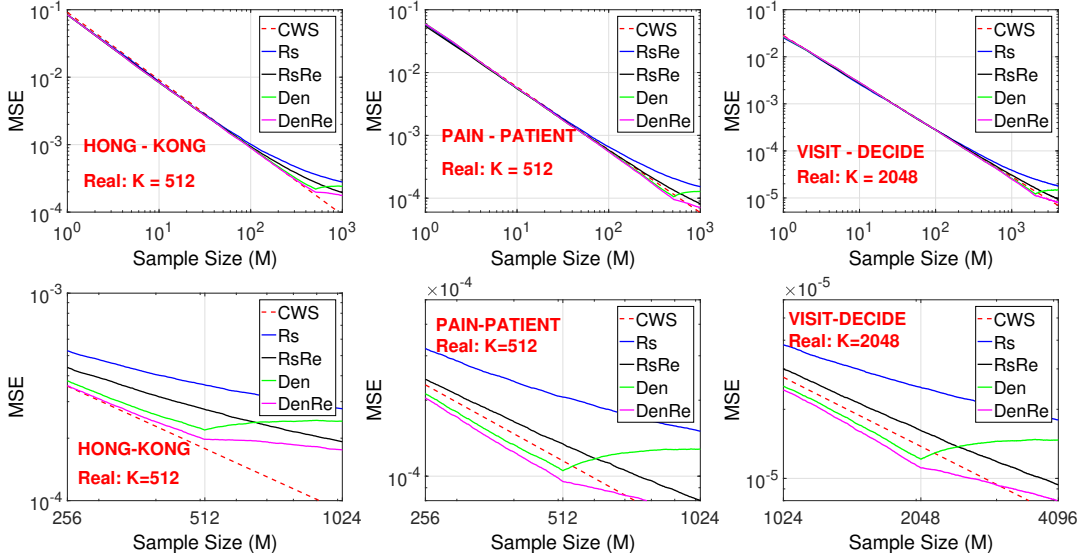

Figure 3: Empirical MSEs of four BCWS schemes for estimating Jaccard similarity on weighted datasets, for three word-vector-pairs. The bottom panels are zoomed-in versions of the upper panels.

similarity $J$ and same variance as CWS. The running time of R-G method is $O(\bar{f} + K\frac{1}{s})$ of a set $S$ with $\bar{f}$ nonzero entries, where $s = \frac{\sum_{i=1}^{D} S_i}{\tilde{m}_D}$ is the effective sparsity. Note that $s = \frac{f}{D}$ when data are binary. Therefore it should be obvious that the R-G algorithm would perform poorly in binary data and also poorly in sparse data.

Recall that, for BCWS using **DenRe** (consider only $M = K$), the running time is $O(\bar{f} + 2K + \frac{K}{K-N_{emp}^K}N_{emp}^K + \frac{\bar{f}}{K}N_{emp}^K)$. Roughly speaking, in a typical situation, we can say that the cost for generating $K$ samples using R-G is $O(K/s)$ and for BCWS is $O(\bar{f})$. Therefore, we can use $\frac{K}{s\bar{f}}$ as an indicator for the improvement of BCWS compared to R-G.

Through a careful study of the literature, the history of the "R-G" algorithm can be traced to [17, 6]. The recent work [29] developed an effective column-wise preprocessing scheme which made the algorithm practical (in dense data). [29] also provided an elegant theoretical analysis to clearly reveal the advantage of the method in dense data (and its disadvantage in sparse data).

In Figure 5, we present the empirical comparisons between R-G and BCWS on two datasets: (i) "Words" ($\frac{1}{s\bar{f}} \approx 1$); (ii) "20 NewsGroup" ($\frac{1}{s\bar{f}} \approx 14 \sim 150$). There is an important (hidden) detail in the R-G algorithm that its performance largely depends on the properly chosen scaling factor. For the original 20 Newsgroup dataset, $\frac{1}{s\bar{f}} \approx 150$, but if we scale the data properly, this value can be reduced substantially to 14. We will explain the R-G algorithm in more details in the supplementary material.

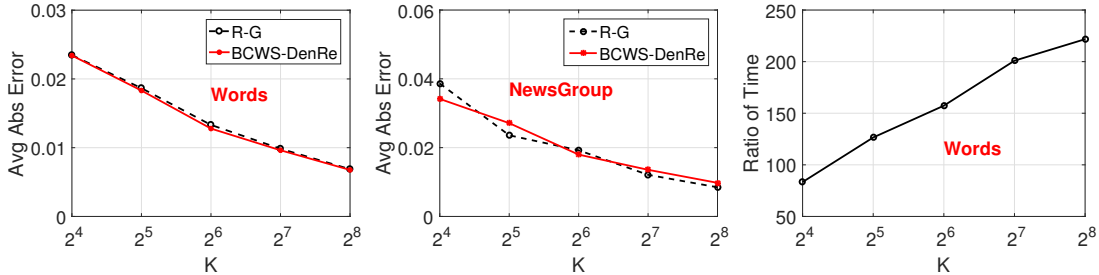

Figure 5: Average absolute estimation error of Jaccard similarity and running time comparison.

The "Words" dataset [23] consists of $2,702$ word-vectors (e.g., "HONG" and "KONG") from a repository of $D = 2^{16}$ documents, for a total of 3,649,051 word-pairs. The left panel of Figure 5 reports the averaged absolute errors (among over 3 million pairs). Similarly we present the errors for 20 NewsGroup, for both R-G and BCWS-DenRe. We also include the time comparisons in the right panel of Figure 5. Basically, for $k = 256$, R-G needs 200 times more time than BCWS. This is largely consistent with what theoretical results would predict. For 20 NewsGroup, we observe that the improvement in efficiency by using BCWS would be even much more substantial.

Finally, we should add that one can combine the idea of BCWS and R-G to significantly speed up the R-G algorithm as suggested in Line 5 of Algorithm 2. We can perhaps name this new method as "B-R-G". Its computational cost would be merely $O\left(\bar{f} + \frac{1}{s}\right)$ as opposed to $O\left(\bar{f} + K\frac{1}{s}\right)$ for R-G. This new method would be very useful for hashing Jaccard similarity in dense high-dimensional data.

### 4.4 Classification Experiment

[20, 21, 22] already conducted extensive experiments on many classification tasks using the min-max kernel (and other kernels) and linearized min-max kernels by CWS hashing.

Here, we report additional experiments on the UCI-Dailysports dataset. When a classifier based on linear SVM is used on the original data, the test accuracy is only $77\%$. However, with the min-max kernel, the accuracy becomes $99\%$. This is a good example to show that min-max kernel (and linearization by CWS hashing) might be very useful in practice. Figure 6 shows that for BCWS with $K \in \{16, 32, 64, 128\}$, using linear SVM and hashed data by BCWS achieves good classification accuracy. Compared to CWS, the accuracy of BCWS is similar or even slightly better. Note that the dimension of the dataset is only 5,625. For example, when $K = 128$ and the desired number of (nonzero) features is $2^{10}$ (x-axis), we have to repeat BCWS $1024/128 = 8$ times. Nevertheless, we can still achieve a cost reduction by a factor of 128 without losing accuracy.

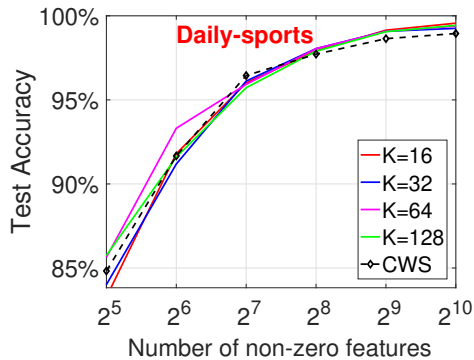

Figure 6: Using linear SVM on the original data achieves only a $77\%$ accuracy. After we hash the data via CWS and use linear SVM on top of hashed data (dashed curve), the accuracy reaches $99\%$. Using BCWS with $K$ ranging from 16 to 128 (solid curves) still attain similar accuracies.

This is a significant part of the contribution in this paper for machine learning. That is, we are able to achieve good accuracy of nonlinear kernels at the cost similar to that of linear classifiers, and the preprocessing (hashing) cost is no longer the bottleneck, unlike the original CWS hashing method.

## 5 Conclusion

We expect BCWS would be adopted in practice for large-scale similarity search and machine learning tasks, given its simplicity and effectiveness. The prior work [20] showed that the min-max kernel, even though it appears simple, could be a good choice of nonlinear kernels for many classification tasks. The more recent work [21] extended the min-max kernel to data vectors with negative entries. In addition, the min-max kernel can be modified to admit tuning parameters [22] for potentially achieving even better performance. The work [22] compared "tunable" min-max kernels with boosted trees and deep nets and presented surprising results. Nevertheless, the processing (hashing) cost of the original CWS algorithm makes it difficult for min-max kernel (and variants) and CWS to be adopted in practice. This study fills in this gap by developing the Bin-Wise CWS (BCWS) algorithm and providing the theoretical analysis. For binary (0/1) data, our results are also interesting and practically useful, in that we provide a scheme that, under the same storage budget, can achieve the provably smallest variance among all existing densification methods for one permutation hashing (OPH).

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
