[Supplementary Material]

# Supplemental Materials for: Re-randomized Densification for One Permutation Hashing and Bin-wise Consistent Weighted Sampling

**Ping Li**
Cognitive Computing Lab
Baidu Research
Bellevue, WA 98004, USA
liping11@baidu.com

**Xiaoyun Li**[*]
Department of Statistics
Rutgers University
Piscataway, NJ 08854, USA
xiaoyun.li@rutgers.edu

**Cun-Hui Zhang**[†]
Department of Statistics
Rutgers University
Piscataway, NJ 08854, USA
cunhui@stat.rutgers.edu

## A  The "Words" Dataset

In this paper, we use the "Words" dataset [4], which consist of 2702 vectors. Each vector is a $2^{16}$-dim sparse vector whose entries are the numbers of occurrences of one word in a repository of $D = 2^{16}$ English documents. For example, the word-vector "HONG" represents the vector of occurrences of word "HONG" in this repository. Note that in Figure 2 of the main paper, we use only the binarized versions of "HONG" and "KONG".

Table 1 lists four pairs of word-vectors in our experiments. They are all fairly sparse (especially the first two pairs). Nevertheless, each vector still has a significant number of nonzero entries. This is quite typical with data in industrial applications, relatively sparse but with a large number of nonzeros in the absolute scale.

Table 1: Statistics of the four pairs of word-vectors used in the experiments.

| Word 1 ($S_1$) | # Word 2 ($S_2$) | # Nonzeros in $S_1$ | # Nonzeros in $S_2$ | Jaccard |
|---|---|---|---|---|
| HONG | KONG | 940 | 948 | 0.8985 |
| PAIN | PATIENT | 886 | 603 | 0.0629 |
| REVIEW | PAPER | 3197 | 1944 | 0.0502 |
| VISIT | DECIDE | 4619 | 954 | 0.0291 |

This motivates us to develop efficient algorithms for consistent weighted sampling (CWS). The processing cost for each vector is $O(\bar{f}K)$, with $\bar{f}$ being the average number of nonzeros and $K$ being the number of samples. Even though in general the data are sparse, the value of $\bar{f}$ can still be large. Ideally, it would be desirable to reduce the cost from $O(fK)$ to just $O(\bar{f})$. Also, note that since we only have to touch each nonzero at least once, this cost $O(\bar{f})$ is minimal and cannot be avoided.

In this paper, we develop Bin-wise consistent weighted sampling (BCWS) with four variants; and we recommend **BCWS-DenRe**. The processing cost for each vector is essentially $O(\bar{f})$ and we have developed the theory to show that the estimates are within a small neighborhood of the true Jaccard similarity. Nevertheless, we would like to present the empirical mean square error (MSE) results to verify that the estimates by BCWS do not deviate much from the truth.

We have conducted a large number of simulations for estimating the Jaccard similarity between two vectors. The patterns are essentially similar and hence we only present the results for four word-vector-pairs, in Figure 1 (which are the enlarged versions of Figure 3 in the main paper). We

---

[*]The work of Xiaoyun Li was conducted during the internship at Baidu Research.
[†]The work of Cun-Hui Zhang was conducted as a consulting researcher at Baidu Research.

can see that for reasonably large $K$ values, BCWS (especially DenRe) methods provide accurate estimates compared to the original CWS method. When the number of samples $M$ is larger than the number of bins $K$, the errors start to be larger than the error of the original CWS. Nevertheless, there is no disastrous effect, especially for DenRe (whose errors continue to decrease even with $M > K$).

Figure 1: Empirical MSEs of four schemes for estimating Jaccard similarity on weighted datasets. The right panels are zoomed-in versions of the left panels.

# B  Double-CWS: An Alternative to BCWS

In retrospect, BCWS-DenRe appears to be a natural idea. Nevertheless, it has taken us quite a journey to eventually obtain this method in the current form.

We would like to comment on a variant of BCWS, which is the initial attempt of this project. As mentioned in the main text, for weighted data, different bins contain different amount of information in nature. Therefore, we tried to select bins according to their importance, in some sense "proportional" to the each bin weight which is the summation of all elements in the bin. Following this direction, an extra CWS sampling has to be implemented on the bin weights for bin selection. This procedure called "double-CWS" creates a densification scheme in nature, and can better handle extreme cases where bin information is highly unbalanced, e.g., data contains large outliers. However, although double-CWS addresses more on bins with large weight, theoretical analysis shows that this procedure tends to scale down the estimates and introduce more bias. Therefore, in general cases, double-CWS is not as good as BCWS. We provide some experimental results in Figure 2 for double-CWS approach.

Figure 2: Empirical MSEs of four schemes for estimating Jaccard similarity on weighted datasets.

# C  Detailed Implementation of the R-G Algorithm

Figure 3:  Illustration of the R-G algorithm in [7]. The green region represents the data entries, and green+red region is the cumulative upper bounds.

In this section we provide more detailed description and analysis of the R-G algorithm, which can be traced to [3, 2] and is developed into the current form due to the effort of [7]. R-G is based on a simple and elegant idea. For a (non-negative) dataset $X \in \mathbb{R}^{n \times D}$, R-G method requires predetermined column-wise upper bound $m_i$, $i = 1, ..., D$, for example, $m_i = \max X_{(:,i)}$, but it can also be set to a larger number to accommodate (e.g.,) future data points. We denote $\tilde{m}_j = \sum_{i=1}^{j} m_i$. For a sample to generate one hash value, in each iteration, the algorithm has two steps:

1. randomly sample a point uniformly on $[0, \tilde{m}_D]$.
2. check if this point falls in green region.

For the first step, one will need to generate and store a large number of random values a priori so the time is ideally negligible. In order to reduce the running time of the second step to $O(1)$, [7] proposes to build a hash map (Algorithm 1) that allows fast search by producing lookup tables. This procedure typically requires $m_i$'s to be integers. Therefore, the column-wise maximal value is set to $\lceil m_i \rceil$. Note that if the data are far away from the integer ceilings, e.g., all at the order of 0.01 (green region), then the algorithm would be very slow, since the red region in this case is large (approximately 1-0.01=0.99). This disadvantage can be ameliorated by re-scaling (e.g., by a factor of 10 or 100).

---

**Algorithm 1:** Compute Hash Map

**Input:** $\tilde{m}_i$, $i = 1, ..., D$.
**Initialize :** index=0,
        CompToM[0]=0
1 **for** $i = 0$ *to* $D - 1$ **do**
2      **if** $i < D - 1$ **then**
3          CompToM[i+1]=$\tilde{m}_i$+CompToM[i]
4      **for** $j = 0$ *to* $\tilde{m}_i$-1 **do**
5          IntToComp[index]=i;
6          index++

**Output:** CompToM[],IntToComp[]

---

The R-G method is an elegant idea. Nevertheless, it exhibits drawbacks which motivate us to develop more efficient algorithms for practical applications.

Firstly, one needs to know the upper bound on each coordinate of the data matrix, which may not be reasonable in some applications. As more and more data points are collected, some values may exceed the maximum set a priori. Therefore, for safety one may have to set a large $m_i$, which would substantially slow down the algorithm.

Secondly, to make R-G work efficiently, one will often have to scale the data with a proper scaling factor. This can also be problematic. For example, the coordinates may not be on same scale. One can choose to scale up largely, e.g. 1,000∼10,000, but this may at the same time lead to a huge hash table.

Lastly, as discussed in the paper [7], the algorithm performs poorly on sparse datasets. However, sparse data are very common in large scale applications, such as text data, web pages, social network and etc.. As a comparison, our proposed BCWS approach is flexible, and can do well in almost all cases provided a suitably chosen number of bins $K$. For example, even with dense data, BCWS improves the speed of the original CWS by a factor $K$.

We re-iterate that the cost of BCWS is essentially $O(\bar{f})$ for processing one vector, where $\bar{f}$ is the average number of nonzeros. Since one will anyway have to touch each data entry once, this cost $O(\bar{f})$ is actually minimal and cannot be avoided even for R-G algorithm. In the main paper, we comment that the ratio of computing times is $O\left(\frac{1}{s\bar{f}}\right)$ (the time of R-G over the time of BCWS). This ratio can be more preciously written as $1 + O\left(\frac{1}{s\bar{f}}\right)$ in the case of dense data.

# D Proofs of Theorems

## D.1 Proof of Lemma 1

*Proof.* **Without conditioning on $m$.** Recall the notation $d = \frac{D}{K}$, and $\tilde{f} = \sum_{i \in I_B} \max(S_i, T_i)$ the number of nonzeros in a bin $B$. To compute the explicit expression of $E_0$, we notice that the number of simultaneously nonzero elements in a bin is given by a hyper-geometric distribution. We have

$$P(\tilde{f} = j) = \frac{\binom{f}{j}\binom{D-f}{d-j}}{\binom{D}{d}}, \quad \max(0, d+f-D) \leq j \leq \min(d, f).$$

We want to compute the expectation conditional on $\tilde{f} \geq 1$. Note that when $d + f - D \geq 1$, we have $P(\tilde{f} = 0) = 0$. Hence the conditional expectation is simply the unconditional one,

$$E_0 = \mathbb{E}[\frac{1}{\tilde{f}} | \tilde{f} \geq 1] = \frac{\sum_{j=\max(0,d+f-D)}^{\min(d,f)} \frac{1}{j}\binom{f}{j}\binom{D-f}{d-j}}{\binom{D}{d}}. \tag{1}$$

When $d + f - D \leq 0$, we have

$$E_0 = \mathbb{E}[\frac{1}{\tilde{f}} | \tilde{f} \geq 1] = \frac{\sum_{j=1}^{\min(d,f)} \frac{1}{j}\binom{f}{j}\binom{D-f}{d-j}}{\binom{D}{d}(1 - P(\tilde{f}=0))} = \frac{\sum_{j=1}^{\min(d,f)} \frac{1}{j}\binom{f}{j}\binom{D-f}{d-j}}{\binom{D}{d}(1 - \frac{\binom{D-f}{d}}{\binom{D}{d}})}$$

$$= \frac{\sum_{j=1}^{\min(d,f)} \frac{1}{j}\binom{f}{j}\binom{D-f}{d-j}}{\binom{D}{d} - \binom{D-f}{d}}. \tag{2}$$

Putting (1) and (2) together gives the result.

**Conditional probability.** To compute $\tilde{E}_0(m)$, it suffices to derive the probability mass function $\mathbb{P}[\tilde{f} = j | I_{emp,k} = 0, m]$, given $m$ non-empty bins. The problem can be formulated as a classical probability model. Consider placing $f$ balls into $K$ bins each with $d$ available positions, without replacement. In our application, since the randomness is brought by permutation and the capacity of each bin is limited, the probability of each ball being placed at each position, instead of in each bin, is uniform. We want to compute the probability distribution of the number of balls in a bin, given that it is one of the $m$ non-empty bins, which is known in priori. Given this condition, we may restrict the analysis in the $m$ non-empty bins which contain all $f$ balls.

First, we calculate the total number of ways to place the balls, such that all $m$ bins are non-empty. Denote $S(k, n|d)$ as the number of ways for placing $n$ balls into $k$ non-empty bins. It is easy to infer the recursion

$$S(k, n|d) = \sum_{j=\max\{1, n-(k-1)d\}}^{\min\{d, n-k+1\}} \binom{d}{j} S(k-1, n-j|d),$$

with boundary condition

$$S(1, n|d) = \binom{d}{n}.$$

For our problem, the number of assignments is thus given by $S(m, f|s)$.

Given a bin $B$, the number of ways to put $j$ balls in $B$ and the rest in the other $(m-1)$ bins such that all bins are non-empty is simply

$$\binom{d}{j} S(m-1, f-j|d).$$

Therefore, we obtain for $\max\{1, f - (m-1)d\} \leq j \leq \min\{d, f-m+1\}$,

$$P[\tilde{f} = j | I_{emp,k} = 0, m] = \frac{\binom{d}{j} S(m-1, f-j|d)}{S(m, f|s)},$$

and $P[\tilde{f} = j | I_{emp,k} = 0, m] = 0$ otherwise. Therefore, we finally derive

$$\tilde{E}_0(m) \triangleq E[\frac{1}{\tilde{f}} | I_{emp,k} = 0, m] = \sum_{j=1}^{d} \frac{1}{j} \mathbb{P}[\tilde{f} = j | I_{emp,k} = 0, m]$$

$$= \sum_{j=\max\{1, f-(m-1)d\}}^{\min\{d, f-m+1\}} \frac{1}{j} \frac{\binom{d}{j} S(m-1, f-j|d)}{S(m, f|s)}.$$

This completes the proof. $\square$

### D.2 Proof of Theorem 1

*Proof.* To proceed with the analysis, first we separate the event of matching hashes into two distinct events. Denote $C_k^E$ the indicator of hash collision at bin $k$ when $k$ is empty, and $C_k^N$ the indicator of collision when $k$ is simultaneously non-empty. Consequently we can write

$$C_k = C_k^N + C_k^E.$$

Recall that $I_{emp,i}$ is the indicator function of the $k$-th bin being empty. According to [5], we have

$$\mathbb{E}(C_k^N | I_{emp,i} = 0) = \mathbb{E}(C_k^E | I_{emp,i} = 0) = J,$$

and also

$$\mathbb{E}[(C_k^N)^2 | I_{emp,i} = 0] = \mathbb{E}[(C_k^E)^2 | I_{emp,i} = 0] = J.$$

Based on above notations, for all schemes we can write

$$\hat{J} = \frac{1}{M} \sum_{i=1}^{M} (C_i^E + C_i^N).$$

Note that here $M$ not necessarily equals to $K$. As all schemes provide unbiased estimator, we can compute

$$Var(\hat{J}) = \mathbb{E}[\frac{1}{M^2} (\sum_{i=1}^{M} (C_i^E + C_i^N))^2] - J^2 \triangleq \frac{1}{M^2} A - J^2. \tag{3}$$

It suffices to analyze $A$. Conditional on the event that the number of non-empty bins $K - N_{emp}^K = m$, we have

$$A = \mathbb{E}[\mathbb{E}[(\sum_{i=1}^{M} (C_i^E + C_i^N))^2 | K - N_{emp}^K = m]]$$

For simplicity we will use $m$ to denote the event $K - N_{emp}^K = m$.

**Rs method.** Recalling the procedure of this method, first we split the set into $K$ bins, and then apply min-hashing for each non-empty bin. We can think these hash values as a fixed table, and we randomly select $M$ hash values from the table. As the permutation and selection are all perfectly random, every selected item should share same property. Hence,

$$A = \mathbb{E}[\mathbb{E}[(\sum_{i=1}^{M} (C_i^E + C_i^N))^2 | m]]$$

$$= \mathbb{E}[\mathbb{E}[\sum_{i=1}^{M} [(C_i^E)^2 + (C_i^N)^2] + \sum_{i \neq j} C_i^E C_j^E + 2 \sum_{i \neq j} C_i^E C_j^N + \sum_{i \neq j} C_i^N C_j^N | m]]$$

$$= \mathbb{E}[\mathbb{E}[\sum_{i=1}^{M} (C_i^E)^2 + \sum_{i \neq j} C_i^E C_j^E | m]] \tag{4}$$

$$= \mathbb{E}[MJ + M(M-1)E_1],$$

where $E_1 = \mathbb{E}[\frac{1}{m} J + (1 - \frac{1}{m}) J\tilde{J}]$. (4) is because we are treating all bins as empty, so the terms involving $C^N$ should all be zero. $E_1$ comes from the fact that if two empty bins $i$ and $j$ choose the

same non-empty bin (with probability $\frac{1}{m}$), then $\mathbb{E}[C_i^E C_j^E | m] = J$. If two distinct bins are chosen, then $\mathbb{E}[C_i^E C_j^E | m] = J\tilde{J}$. Replacing $m = K - N_{emp}^K$ yields

$$Var_{Rs}(M) = \frac{J}{M} + \frac{M-1}{M}E_1 - J^2.$$

**RsRe method.** Let $B$ be a non-empty bin, and $h_1 = (h_1^S(B), h_1^T(B))$ is a pair of hash values generated for sets $S$ and $T$. Let $h_2 = (h_2^S(B)$ and $h_2^T(B))$ be another pair of hashes generated independently from the same $B$. By the property of one permutation hashing and independence among different hash samples, the collision probability within a non-empty bin is unbiased in the sense that

$$P(h_1^S(B) = h_1^T(B)|I_{emp,B} \neq 0) = P(h_2^S(B) = h_2^T(B)|I_{emp,B} \neq 0) = J. \qquad (5)$$

Denote the left-hand side of (5) as $Pr(\Omega)$. Consider two permutations on $\pi_1$ and $\pi_2$ on $\mathcal{D} = \{1, ..., D\}$. Denote $m_1 = \min(\pi_1)$ and $m_2 = \min(\pi_2)$. We have

$$\begin{aligned} P(\Omega|I_{emp,B} \neq 0) = & P(\Omega|I_{emp,B} \neq 0, m_1 = m_2)P(m_1 = m_2|I_{emp,B} \neq 0) \\ & + P(\Omega|I_{emp,B} \neq 0, m_1 \neq m_2)P(m_1 \neq m_2|I_{emp,B} \neq 0). \end{aligned} \qquad (6)$$

This expression helps us to gain some insight on the randomness brought by re-randomization. If two permutations $\pi_1$ and $\pi_2$ happen to give same minimum index, $i.e$ $\min(\pi_1) = \min(\pi_2) = i$ for some index $i$, then actually two pairs of hashes $h_1$ and $h_2$ only contains the information of one pair of samples $(S_i, T_i)$. We can understand this argument as forcing $h_1$ and $h_2$ to be equal. Hence,

$$P(\Omega|I_{emp,B} \neq 0, m_1 = m_2) = J. \qquad (7)$$

Similarly, if $\min(\pi_1) \neq \min(\pi_2)$, $h_1$ and $h_2$ will contain two pieces of information. Precisely, we have

$$P(\Omega|I_{emp,B} \neq 0, m_1 \neq m_2) = J\tilde{J}. \qquad (8)$$

Moreover, since the permutations are perfectly random, the probability of $\min(\pi_1) = \min(\pi_2) = i$ will depend on the number of simultaneously nonzero elements in bin $B$. We have, given the number of non-empty bins $m$,

$$P(m_1 = m_2|I_{emp,B} \neq 0, m) = \mathbb{E}[\frac{1}{\tilde{f}}|I_{emp,B} \neq 0, m], \qquad (9)$$

where $\tilde{f} = \sum_{i \in B} max(S_i, T_i)$. From above analysis, we have for $\forall i \neq j$,

$$\begin{aligned} \mathbb{E}[C_i^E C_j^E | m] &= \mathbb{P}[C_i^E = C_j^E = 1|m] \\ &= \frac{1}{m}\mathbb{P}[C_i^E = C_j^E = 1|\sigma(i) = \sigma(j), m] + \frac{m-1}{m}\mathbb{P}[C_i^E = C_j^E = 1|\sigma(i) \neq \sigma(j), m] \\ &= \frac{1}{m}\mathbb{P}[h_1^v(B_k) = h_1^w(B_k), h_2^v(B_k) = h_2^w(B_k)|I_{emp,k} = 0, m] + \frac{m-1}{m}J\tilde{J} \\ &= \frac{1}{m}\left[J\mathbb{E}[\frac{1}{\tilde{f}}|I_{emp,k} = 0, m] + J\tilde{J}(1 - \mathbb{E}[\frac{1}{\tilde{f}}|I_{emp,k} = 0, m])\right] + \frac{m-1}{m}J\tilde{J} \\ &= \frac{1}{m}\left[J\tilde{E}_0(m) + J\tilde{J}(1 - \tilde{E}_0(m))\right] + \frac{m-1}{m}J\tilde{J}, \end{aligned}$$

where $E_0(m)$ is defined in Lemma 1. Taking expectation w.r.t. $m$, we get

$$\mathbb{E}[\mathbb{E}[\sum_{i \neq j} C_i^E C_j^E | m]] = M(M-1)E_2,$$

with

$$E_2 = \mathbb{E}[\frac{\tilde{E}_0(m)J + (1 - \tilde{E}_0(m))J\tilde{J}}{m} + \frac{m-1}{m}J\tilde{J}] = \mathbb{E}[\frac{\tilde{E}_0(m)}{m}J + (1 - \frac{\tilde{E}_0(m)}{m})J\tilde{J}].$$

Replacing $m$ by $(K - N_{emp}^K)$ in (3) gives

$$Var_{RsRe}(M) = \frac{J}{M} + \frac{M-1}{M}E_2 - J^2.$$

**Den method.** When $M \leq K$, we have

$$A = \mathbb{E}[\mathbb{E}[(\sum_{i=1}^{M}(C_i^E + C_i^N))^2|m]]$$

$$= \mathbb{E}[\mathbb{E}[\sum_{i=1}^{M}[(C_i^E)^2 + (C_i^N)^2] + \sum_{i \neq j}C_i^E C_j^E + 2\sum_{i \neq j}C_i^E C_j^N + \sum_{i \neq j}C_i^N C_j^N|m]]$$

$$= \mathbb{E}[MJ + N_{emp}^M(N_{emp}^M - 1)E_1 + 2N_{emp}^M(M - N_{emp}^M)E_1$$
$$+ (M - N_{emp}^M)(M - N_{emp}^M - 1)J\tilde{J}]$$

$$= \mathbb{E}[MJ + N_{emp}^M(2M - N_{emp}^M - 1)]E_1 \tag{10}$$
$$+ (M - N_{emp}^M)(M - N_{emp}^M - 1)J\tilde{J}.$$

Combining with (3) yields

$$Var_{Den}(M) = \frac{J}{M} + \frac{1}{M^2}\mathbb{E}[(M - N_{emp}^M)(M - N_{emp}^M - 1)J\tilde{J}]$$
$$+ \frac{1}{M^2}\mathbb{E}[N_{emp}^M(2M - N_{emp}^M - 1)]E_1 - J^2. \tag{11}$$

When $M > K$, the first $K$ bins are the $K$ bins from original split, and last $(M - K)$ bins can be regarded as extra empty bins. Denote $H_i^E$ the indicator of event that matching hash values occur at bin $i$, with $i > K$. We modify (3) to

$$Var_{Den}(\hat{J}) = \mathbb{E}[\frac{1}{M^2}((\sum_{i=1}^{K}C_i^E + C_i^N) + \sum_{i=K+1}^{M}H_i^E)^2] - J^2 \triangleq \frac{1}{M^2}A^+ - J^2, \tag{12}$$

and

$$A^+ = (\sum_{i=1}^{K}(C_i^E + C_i^N))^2 + 2(\sum_{i=1}^{K}(C_i^E + C_i^N)) \cdot \sum_{i=K+1}^{M}H_i^E + (\sum_{i=K+1}^{M}H_i^E)^2. \tag{13}$$

For the first part of $A^+$, we have

$$\mathbb{E}[(\sum_{i=1}^{K}(C_i^E + C_i^N))^2] = \mathbb{E}[[(\sum_{i=1}^{K}(C_i^E + C_i^N))^2|m]] = K^2(V_{Den}(K) + J^2). \tag{14}$$

In addition,

$$\mathbb{E}[[\sum_{i=1}^{K}C_k^E \cdot \sum_{i=K+1}^{M}H_i^E|m]] = N_{emp}^K(M - K)[\frac{1}{m}J + (1 - \frac{1}{m})J\tilde{J}]$$
$$= N_{emp}^K(M - K)E_1, \tag{15}$$

$$\mathbb{E}[[\sum_{i=1}^{K}C_i^N \cdot \sum_{j=K+1}^{M}H_j^E|m]] = (K - N_{emp}^K)(M - K)[\frac{1}{m}J + (1 - \frac{1}{m})J\tilde{J}]$$
$$= (K - N_{emp}^K)(M - K)E_1, \tag{16}$$

$$\mathbb{E}[[(\sum_{i=K+1}^{M}H_i^E)^2] = (M - K)(M - K - 1)[\frac{1}{m}J + (1 - \frac{1}{m})J\tilde{J}] + (M - K)J$$
$$= (M - K - 1)(M - K)E_1 + (M - K)J. \tag{17}$$

Combining (11)-(17) together we obtain for $M > K$,

$$V_{Den}(M) = \frac{1}{M^2}[K^2(V_{Den}(K) + J^2) + (M - K)(M + K - 1)E_1 + (M - K)J] - J^2.$$

**DenRe method.** The proof procedures for **Den** scheme also hold in this case. We just need to replace $E_1$ to $E_2$ and everything else will follow. $\qquad\square$

### D.3 Proof of Theorem 2

*Proof.* Suppose $D$ is a multiplier of $K$ and $D$ cells are divided into $K$ bins of equal size. Suppose $f$ out of the $D$ cells are assigned value 1 and rest value 0. Let $I_{emp,i}$ be the indicator that all cells in the $i$-th bin are assigned 0. Due to the exchangeability of $I_{emp,i}$,

$$P\Big\{\sum_{i=1}^{M} I_{emp,i} = j\Big\} = \binom{M}{j}\Big[\Big(\prod_{i=1}^{j} I_{emp,i}\Big)\Big\{\prod_{i=j+1}^{M}\big(1 - I_{emp,i}\big)\Big\}\Big]$$

$$= \binom{M}{j}\Big[\Big(\prod_{i=1}^{j} I_{emp,i}\Big)\Big\{1 + \sum_{\ell=1}^{M-j}(-1)^{\ell}\binom{M-j}{\ell}\prod_{i=j+1}^{j+\ell} I_{emp,i}\Big\}\Big]$$

$$= \sum_{\ell=0}^{M-j}(-1)^{\ell}\binom{M}{j}\binom{-j}{\ell}\Big(\prod_{i=1}^{j+\ell} I_{emp,i}\Big).$$

This is due to the generalized inclusion-exclusion formula. Thus, for $M \leq K$,

$$P\Big\{\sum_{i=1}^{M} I_{emp,i} = j\Big\} = \sum_{\ell=0}^{M-j}(-1)^{\ell}\binom{M}{j}\binom{M-j}{\ell}\binom{D(1-(j+\ell)/K)}{f}\Big/\binom{D}{f}.$$

□

### D.4 Proof of Theorem 4

The following lemma [1, 6] is useful for the analysis.

**Lemma 2.** *Let $\{a_1, ..., a_n\}$ be a finite population with $\bar{a} = \frac{\sum_{i=1}^{n} a_i}{n}$ and $\sigma^2 = \frac{\sum_{i=1}^{n}(a_i - \bar{a})^2}{n}$. Let $A$ be a random subset of $\{1, ..., n\}$ of deterministic size $|A| = n_A$. Define $\bar{a}_A = \frac{\sum_{i \in A} a_i}{n}$ and $p_A = \frac{n_A}{n}$. Then we have for all $t > 0$,*

$$P\{n_A(\bar{a}_A - \bar{a}) > (1 + \epsilon_0)\sigma n t\} \leq e^{-nt^2}, \tag{18}$$

*where $\epsilon_0 \leq \min\{\frac{1}{70}, \frac{9p_A^2}{70}, \frac{9(1-p_A)^2}{70}\}$. When $\frac{n}{n_A}$ is an integer, we have $\epsilon_0 = 0$.*

**Proof of Theorem 4.**

*Proof.* Let $I_k$ be the collection of elements in bin $k$. First we notice that

$$\mathbb{E}[\hat{J}_{BCWS}(\pi)] = \frac{1}{K}\sum_{k=1}^{K} J_k(\pi),$$

where $J_k(\pi) = \frac{\sum_{i \in I_k} S_i \wedge T_i}{\sum_{i \in I_k} S_i \vee T_i}$ is the Jaccard similarity in the $k$-th bin. Note that all the bins are non-empty under DenRe scheme. By Lemma 2, we have

$$P\Big\{\Big|\sum_{i \in I_k}\frac{S_i}{|I_k|} - \mu_1\Big| \geq \frac{D\sigma_1}{|I_k|}\sqrt{\frac{t}{D}}\Big\} \leq 2e^{-t},$$

$$P\Big\{\Big|\sum_{i \in I_k}\frac{T_i}{|I_k|} - \mu_2\Big| \geq \frac{D\sigma_2}{|I_k|}\sqrt{\frac{t}{D}}\Big\} \leq 2e^{-t},$$

$$P\Big\{\Big|\sum_{i \in I_k}\frac{S_i \vee T_i}{|I_k|} - \mu_3\Big| \geq \frac{D\sigma_3}{|I_k|}\sqrt{\frac{t}{D}}\Big\} \leq 2e^{-t}.$$

The $\epsilon_0$ term is gone because $\frac{D}{|I_k|} = K$ is an integer. Denote the event $\Theta = \{|\sum_{i \in I_k} \frac{S_i}{|I_k|} - \mu_1| \geq \frac{D\sigma_1}{|I_k|}\sqrt{\frac{t}{D}}\}$. Now we have

$$P\{\Theta|I_{emp,k}\} = \frac{P\{\Theta, I_{emp,k} = 0\}}{P[I_{emp,k} = 0]} = \frac{1}{P[I_{emp,k} = 0]}\{P(\Theta) - P(\Theta|I_{emp,k} = 1)P[I_{emp,k} = 1]\}$$

$$\leq \frac{2e^{-t}}{P[I_{emp,k} = 0]} - \frac{P[I_{emp,k} = 1]}{P[I_{emp,k} = 0]}$$

$$= \frac{2e^{-t}}{p_1} - \frac{p_0}{p_1},$$

where $p_0 = P[I_{emp,k} = 1] = \binom{D-f}{D/K}/\binom{D}{D/K}$ and $p_1 = 1 - p_0$. The second line is because conditional on $I_{emp,k} = 1$ (bin $k$ is empty), by assumption we have

$$P[\Theta] = P[\mu_1 \geq K\sigma_1\sqrt{\frac{t}{D}}] = P[K \leq \frac{\mu_1}{\sigma_1}\sqrt{\frac{D}{t}}] = 1.$$

Similar arguments hold for $T$ and $S \vee T$. Using union bound, with probability at least $1 - \frac{6K}{p_1}e^{-t} + \frac{3p_0 K}{p_1}$, we have for $\forall k$,

$$\begin{cases} |\sum_{i \in I_k} \frac{S_i}{|I_k|\mu_1} - 1| \leq K\frac{\sigma_1}{\mu_1}\sqrt{\frac{t}{D}} \triangleq K\delta_1(t), \\ |\sum_{i \in I_k} \frac{T_i}{|I_k|\mu_2} - 1| \leq K\frac{\sigma_2}{\mu_2}\sqrt{\frac{t}{D}} \triangleq K\delta_2(t), \\ |\sum_{i \in I_k} \frac{S_i \vee T_i}{|I_k|\mu_3} - 1| \leq K\frac{\sigma_3}{\mu_3}\sqrt{\frac{t}{D}} \triangleq K\delta_3(t). \end{cases} \tag{19}$$

Note that $\sum_{i \in A} S_i \wedge T_i = \sum_{i \in A} S_i + \sum_{i \in A} T_i - \sum_{i \in A} S_i \vee T_i$ always holds for any $A \subseteq \{1, ..., D\}$. Hence we have

$$J = \frac{\mu_1 + \mu_2}{\mu_3} - 1 \tag{20}$$

Therefore, with probability at least $1 - \frac{6K}{p_1}e^{-t} + \frac{3p_0 K}{p_1}$, we have

$$\mathbb{E}[\hat{J}_{BCWS}(\pi)] = \frac{1}{K}\sum_{i=1}^{K} J_k(\pi) = \frac{1}{K}\sum_{i=1}^{K} \frac{\sum_{i \in I_k} S_i + \sum_{i \in I_k} T_i}{\sum_{i \in I_k} S_i \wedge T_i} - 1 \tag{21}$$

$$\geq \frac{|I_k|\mu_1(1 - K\delta_1(t)) + |I_k|\mu_2(1 - K\delta_2(t))}{|I_k|\mu_3(1 + K\delta_3(t))} - 1,$$

$$\geq \frac{(\mu_1 + \mu_2)\min_{i=1,2}(1 - K\delta_i(t))}{\mu_3(1 + K\delta_3(t))} - 1,$$

$$= (J + 1)\frac{1 - K(\delta_1 \vee \delta_2)}{1 + K\delta_3} - 1,$$

$$= \frac{1 - K(\delta_1(t) \vee \delta_2(t))}{1 + K\delta_3(t)}J - \frac{K(\delta_3(t) + (\delta_1(t) \vee \delta_2(t)))}{1 + K\delta_3(t)}. \tag{22}$$

Using similar argument, we also have

$$\mathbb{E}[\hat{J}_{BCWS}(\pi)] \leq \frac{|I_k|\mu_1(1 + K\delta_1(t)) + |I_k|\mu_2(1 + K\delta_2(t))}{|I_k|\mu_3(1 - K\delta_3(t))} - 1,$$

$$\leq \frac{(\mu_1 + \mu_2)\max_{i=1,2}(1 + K\delta_i(t))}{\mu_3(1 - K\delta_3(t))} - 1,$$

$$= (J + 1)\frac{1 + K(\delta_1 \vee \delta_2)}{1 - K\delta_3} - 1,$$

$$= \frac{1 + K(\delta_1(t) \vee \delta_2(t))}{1 - K\delta_3(t)}J + \frac{K(\delta_3(t) + (\delta_1(t) \vee \delta_2(t)))}{1 - K\delta_3(t)}. \tag{23}$$

Combining (22) and (23) completes the proof. $\qquad\square$