[Reviews · NeurIPS 2019]

Reviewer 1



This paper proposed a new rerandomized densification strategy for minhash as well as a bin-wise CWS. The proposed method achieves the smallest variance among all densification schemes. Densification of one permutation hashing is a very important problem, and this work proposed a new technique that achieves the smallest variance. The idea of this paper is quite interesting to me, and I like the analysis and experiment of this paper. Overall, the idea of this paper is interesting and it is a great incremental advance in the field of densification one permutation hashing. So, I would like to give a positive vote for this paper.

Reviewer 2



In this paper, the authors continue the line of work that replaces k independent minhashes, sample only one permutation and then extract k samples for it so to obtain an unbiased estimator for the Jaccard similarity. The authors show a minor variant of the algorithm from [23], where more randomness is injected. They analyze the variance of estimator and conclude that the variance is smaller than for the scheme from [23], as well as evaluate it experimentally. I like the result, since it makes progress on an important problem (estimators for the Jaccard similarity are heavily used in practice). However, the result is a bit incremental, and not really about learning (sure, minhash and its variants can be used for kernel methods, but it's orthogonal to the contribution of the present paper). These considerations make me put this paper into the "slightly above borderline" pile.

Reviewer 3



The authors propose that the optimal densification for OPH can actually be further optimized. In usual OPH, we get one permutation of the sparse vector, break the vector into K equal sized bins. In the usual Consistent Weighted Sampling (CWS) approach, we sample non-empty bins from these K bins and retrieve a fixed hash code for these bins. In this new approach, the authors suggest to treat each of the K bins as a separate sparse vector and perform MinHash on these retrieved bins to get a hash code instead of directly getting a Hash code. This is called re-randomization. The authors theoretically prove that this re-randomization achieves the smallest variance among densification schemes(that are used to retrieve hash codes from empty buckets). Also, they extend this idea to weighted non-negative sparse vectors (by a method called Bin-wise CWS) The paper seems to be a subtle improvement over prior work. It's fairly well written barring some typos (listed below). It is a significant observation but not entirely new. Typos: 1. well behavored ---> well behaved (in multiple places) 2. In section 2, the sentence "we use in J(S,T) to denote Jaccard similarity and J(S,T) to denote generalized Jaccard similarity" sounds weird. Are both denoted by J(S,T)? 3. In section 3, I_s1 = {1,2,3} is wrong. It should be I_s1 = {2,3,4} <===================== POST AUTHOR RESPONSE=================> I've read the author response. I'm still positive abt the submission and I'm retaining my score.

[Author Response · NeurIPS 2019]

# 1 Response to Reviewer #1

Thanks very much for the encouraging feedback and for providing a good summary of our contributions.

We would like to bring to your further attention that, the proposed bin-wise CWS (BCWS) method for non-binary data, which now appears to be quite natural in this paper, was actually not straightforward to develop. In the supplementary material, Section B proposes another strategy – Double-CWS. That is, we first use CWS to choose the bin and then conduct CWS within the chosen bin to generate the sample. Double-CWS was our initial proposal and we had invested significant efforts on the analysis and refinement of the algorithm. Unfortunately, as shown in Figure 2 in the supplementary material, Double-CWS is significantly worse than BCWS, although Double-CWS works reasonably well when sample size is smaller than (e.g.,) 100. The finding our BCWS in the current form was, to be honest, due to an incidental empirical finding. Once we were impressed by its performance in experiments, we were able to derive the concentration bound of BCWS in Theorem 4. Then, the connection to densification schemes becomes obvious.

When applying CWS to real-world machine learning applications, the data processing time, especially the time during testing, is actually the bottleneck. We are glad to see the good empirical performance of BCWS and its substantial speed-up compared to the original expensive CWS algorithm. Thanks again.

# 2 Response to Reviewer #2

Thank you for the positive comments. Yes, as you summarized, our work consists of

1. A variant of densified one permutation hashing (OPH) for binary (0/1) data, which further reduces the estimation variance compared to the elegant work of [23] (which was previously believed to be optimal). This finding and subsequent analysis are useful not only theoretically but also practically.

2. Perhaps surprisingly, densification of OPH for binarry data actually provides a crucial strategy for choosing the bins for our proposed binw-wise CWS (BCWS) for non-binary data. This strategy, which now may appear natural, was actually not so straightforward to derive. In Section B of the supplementary material, we compared BCWS with another "seemingly more obvious" strategy named "Double-CWS" (that is, we first use CWS to choose the bin then apply CWS within the chosen bin to obtain an sample). We can see from Figure 2 in the supplementary material that BCWS substantially improves Double-CWS.

3. We show empirically that BCWS achieved substantial speed-ups compared to the original CWS and the newly proposed "R-G" algorithm [22] for non-binary (sparse) data. We will further elaborate on this point.

There has been good evidence that the Jaccard similarity for non-binary data can achieve good empirical performance in machine learning tasks such as classification and near-neighbor search. In order to use Jaccard similarity efficiently, one will typically have to resort to CWS algorithm to generate samples in order to "linearize" this nonlinear kernel. This CWS "hashing" procedure is actually very time-consuming. Training for massive data will be slow since one has to generate samples many times. More importantly, testing speed might become the bottleneck if one has to generate many CWS samples for newly arrive data vectors. This seriously limits the practical use of CWS and Jaccard kernel.

Therefore, the significance of this submission in the context of machine learning is that it enables a potentially useful kernel to be practical. Similar significance can be concluded in the context of approximate near neighbor search.

# 3 Response to Reviewer #3

Thank you very much for your valuable comments and thanks for kindly pointing the typos.

It is very nice of you to comment that "It is a significant observation". In the rebuttals to two other reviewers, we have partially explained our journey how we derived Bin-wise CWS (BCWS) for non-binary data and its connection to optimal densification for binary data. We hope Section B in the supplementary material may interest you, which presented our initial idea of "Double-CWS". We also hope the comparison with the nice recent work on "R-G" algorithm [22] for sampling Jaccard kernel might be interesting to you. Overall, we now have a practical algorithm which we hope to be able to help practitioners. Thanks again for the encouraging comments and precise summary of our contributions.

[Meta-Review · NeurIPS 2019]

Overall the reviewers appreciated the idea, which although incremental, was quite subtle and interesting, and given also the importance of the problem, this was enough for the reviewers to all agree to push this paper over the bar.